# WELL-READ STUDENTS LEARN BETTER: ON THE IMPORTANCE OF PRE-TRAINING COMPACT MODELS

## ABSTRACT

Recent developments in natural language representations have been accompanied by large and expensive models that leverage vast amounts of general-domain text through self-supervised pre-training. Due to the cost of applying such models to down-stream tasks, several model compression techniques on pre-trained language representations have been proposed (Sun et al., 2019a; Sanh, 2019). However, surprisingly, the simple baseline of just pre-training and fine-tuning compact models has been overlooked. In this paper, we first show that pre-training remains important in the context of smaller architectures, and fine-tuning pre-trained compact models can be competitive to more elaborate methods proposed in concurrent work. Starting with pre-trained compact models, we then explore transferring task knowledge from large fine-tuned models through standard knowledge distillation. The resulting simple, yet effective and general algorithm, *Pre-trained Distillation*, brings further improvements. Through extensive experiments, we more generally explore the interaction between pre-training and distillation under two variables that have been under-studied: model size and properties of unlabeled task data. One surprising observation is that they have a compound effect even when sequentially applied on the *same* data. To accelerate future research, we will make our 24 pre-trained miniature BERT models publicly available.

## 1 INTRODUCTION

Self-supervised learning on a general-domain text corpus followed by end-task learning is the two-staged training approach that enabled deep-and-wide Transformer-based networks (Vaswani et al., 2017) to advance language understanding (Devlin et al., 2018; Yang et al., 2019b; Sun et al., 2019b; Liu et al., 2019). However, state-of-the-art models have hundreds of millions of parameters, incurring a high computational cost. Our goal is to realize their gains under a restricted memory and latency budget. We seek a training method that is *well-performing*, *general* and *simple* and can leverage additional resources such as unlabeled task data.

Before considering compression techniques, we start with the following research question: *Could we directly train small models using the same two-staged approach?* In other words, we explore the idea of applying language model (LM) pre-training and task fine-tuning to compact architectures directly. This simple baseline has so far been overlooked by the NLP community, potentially based on an underlying assumption that the limited capacity of compact models is capitalized better when focusing on the end task rather than a general language model objective. Concurrent work to ours proposes variations of the standard pre-training+fine-tuning procedure, but with limited generality (Sun et al., 2019a; Sanh, 2019). We make the surprising finding that pre-training+fine-tuning in its original formulation is a competitive method for building compact models.

For further gains, we additionally leverage knowledge distillation (Hinton et al., 2015), the standard technique for model compression. A compact student is trained to recover the predictions of a highly accurate teacher. In addition to the posited regularization effect of these *soft* labels (Hinton et al., 2015), distillation provides a means of producing pseudo-labels for unlabeled data. By regarding LM pre-training of compact models as a student initialization strategy, we can take advantage of both methods. The resulting algorithm is a sequence of three standard training operations: masked LM (MLM) pre-training (Devlin et al., 2018), task-specific distillation, and optional fine-tuning. From here on, we will refer to it as *Pre-trained Distillation* (PD) (Figure 1). As we will show in

**Algorithm 1**

**Require:** student $\theta$, teacher $\Omega$, unlabeled LM data $\mathcal{D}_{LM}$, unlabeled transfer data $\mathcal{D}_T$, labeled data $\mathcal{D}_L$

1: Initialize $\theta$ by pre-training an MLM$^+$ on $\mathcal{D}_{LM}$
2: **for each** $x \in \mathcal{D}_T$ **do**
3:     Get loss $L \leftarrow -\sum_y P_\Omega(y|x) \log P_\theta(y|x)$
4:     Update student $\theta \leftarrow \text{BACKPROP}(L, \theta)$
5: **end for**
6: Fine-tune $\theta$ on $\mathcal{D}_L$          ▷ Optional step.
7: **return** $\theta$

Figure 1: Pre-trained Distillation

Section 6.2, PD outperforms the pre-training+fine-tuning (PF) baseline, especially in the presence of a large transfer set for distillation.

In a controlled study following data and model architecture settings in concurrent work (Section 4), we show that Pre-trained Distillation outperforms or is competitive with more elaborate approaches which use either more sophisticated distillation of task knowledge (Sun et al., 2019a) or more sophisticated pre-training from unlabeled text (Sanh, 2019). The former distill task knowledge from intermediate teacher activations, starting with a heuristically initialized student. The latter fine-tune a compact model that is pre-trained on unlabeled text with the help of a larger LM teacher.

One of the most noteworthy contributions of our paper are the extensive experiments that examine how Pre-trained Distillation and its baselines perform under various conditions. We investigate two axes that have been under-studied in previous work: model size and amount/quality of unlabeled data. While experimenting with 24 models of various sizes (4m to 110m parameters) and depth/width trade-offs, we observe that pre-trained students can leverage depth much better than width; in contrast, this property is not visible for randomly-initialized models. For the second axis, we vary the amount of unlabeled data, as well as its similarity to the labeled set. Interestingly, Pre-trained Distillation is more robust to these variations in the transfer set than standard distillation.

Finally, in order to gain insight into the interaction between LM pre-training and task-specific distillation, we sequentially apply these operations on the *same* dataset. In this experiment, chaining the two operations performs better than any one of them applied in isolation, despite the fact that a single dataset was used for both steps. This compounding effect is surprising, indicating that pre-training and distillation are learning complementary aspects of the data.

Given the effectiveness of LM pre-training on compact architectures, we will make our 24 pre-trained miniature BERT models publicly available in order to accelerate future research.

## 2 PROBLEM STATEMENT

Our high-level goal is to build accurate models which fit a given memory and latency budget. There are many aspects to explore: the parametric form of the compact model (architecture, number of parameters, trade-off between number of hidden layers and embedding size), the training data (size, distribution, presence or absence of labels, training objective), etc. Since an exhaustive search over this space is impractical, we fix the model architecture to bidirectional Transformers, known to be suitable for a wide range of NLP tasks (Vaswani et al., 2017; Devlin et al., 2018). The rest of this section elaborates on the training resources we assume to have at our disposal.

**The teacher** is a highly accurate but large model for an end task, that does not meet the resource constraints. Prior work on distillation often makes use of an ensemble of networks (Hinton et al., 2015). For faster experimentation, we use a single teacher, without making a statement about the best architectural choice. In Section 4, the teacher is pre-trained BERT$_{\text{BASE}}$ fine-tuned on labeled end-task data. In Section 6, we use BERT$_{\text{LARGE}}$ instead.

**Students** are compact models that satisfy resource constraints. Since model size qualifiers are relative (e.g., what is considered small in a data center can be impractically large on a mobile device),

|       | H=128 | H=256 | H=512 | H=768 |
|-------|-------|-------|-------|-------|
| L=2   | 4.4  | 9.7  | 22.8 | 39.2  |
| L=4   | 4.8   | 11.3 | 29.1 | 53.4  |
| L=6   | 5.2   | 12.8 | 35.4 | 67.5  |
| L=8   | 5.6   | 14.4 | 41.7 | 81.7  |
| L=10  | 6.0   | 16.0 | 48.0 | 95.9  |
| L=12  | 6.4   | 17.6 | 54.3 | 110.1 |

|       | H=128 | H=256 | H=512 | H=768 |
|-------|-------|-------|-------|-------|
| L=2   | 65.24 | 31.25 | 14.44 | 7.46 |
| L=4   | 32.37 | 15.96 | 7.27 | 3.75 |
| L=6   | 21.87 | 10.67 | 4.85 | 2.50 |
| L=8   | 16.42 | 8.01 | 3.64 | 1.88 |
| L=10  | 13.05 | 6.37 | 2.90 | 1.50 |
| L=12  | 11.02 | 5.35 | 2.43 | 1.25 |

(a) Millions of parameters      (b) Relative speedup wrt $BERT_{LARGE}$ on TPU v2

Table 1: Student models with various numbers of transformer layers (**L**) and hidden embedding sizes (**H**). Latency is computed on Cloud TPUs with batch size 1 and sequence length 512. For readability, we focus on five models (underlined) for most figures: Tiny (L=2, H=128), Mini (L=4, H=256), Small (L=4, H=512), Medium (L=8, H=512), and Base (L=12, H=768).

we investigate an array of 24 model sizes, from our $Transformer_{TINY}$ (4m parameters) all the way up to $Transformer_{BASE}$ (110m parameters)[1]. The student model sizes and their relative speed-up compared to the $BERT_{LARGE}$ teacher can be found in Table 1. Interested readers can situate themselves on this spectrum based on their resource constraints. For readability, most plots show a selection of 5 models, but we verify that our conclusions hold for all 24.

**Labeled data** ($\mathcal{D}_L$) is a set of $N$ training examples $\{(x_1, y_1), ..., (x_N, y_N)\}$, where $x_i$ is an input and $y_i$ is a label. For most NLP tasks, labeled sets are hard to produce and thus restricted in size.

**Unlabeled transfer data** ($\mathcal{D}_T$) is a set of $M$ input examples of the form $\{x'_1, ..., x'_M\}$ sampled from a distribution that is similar to but possibly not identical to the input distribution of the labeled set. During distillation, the teacher *transfers* knowledge to the student by exposing its label predictions for instances $x'_m$. $\mathcal{D}_T$ can also include the input portion of labeled data $\mathcal{D}_L$ instances. Due to the lack of true labels, such sets are generally easier to produce and consequently larger than labeled ones. Note, however, that task-relevant input text is not readily available for key tasks requiring paired texts such as natural language inference and question answering, as well as domain-specific dialog understanding. In addition, for deployed systems, input data distribution shifts over time and existing unlabeled data becomes stale (Kim et al., 2017).

**Unlabeled language model data** ($\mathcal{D}_{LM}$) is a collection of natural language texts that enable unsupervised learning of text representations. We use it for unsupervised pre-training with a masked language model objective (Devlin et al., 2018). Because no labels are needed and strong domain similarity is not required, these corpora are often vast, containing thousands of millions of words.

The distinction between the three types of datasets is strictly functional. Note they are not necessarily disjunct. For instance, the same corpus that forms the labeled data can also be part of the unlabeled transfer set, after its labels are discarded. Similarly, corpora that are included in the transfer set can also be used as unlabeled LM data.

## 3 PRE-TRAINED DISTILLATION

*Pre-trained Distillation* (PD) (Figure 1) is a general, yet simple algorithm for building compact models that can leverage all the resources enumerated in Section 2. It consists of a sequence of three standard training operations that can be applied to any choice of architecture:

1. **Pre-training on** $\mathcal{D}_{LM}$. A compact model is trained with a masked LM objective (Devlin et al., 2018), capturing linguistic phenomena from a large corpus of natural language texts.

2. **Distillation on** $\mathcal{D}_T$. This *well-read* student is now prepared to take full advantage of the teacher expertise, and is trained on the *soft* labels (predictive distribution) produced by the teacher. As we will show in Section 6.2, randomly initialized distillation is constrained by the size and distribution of its unlabeled transfer set. However, the previous pre-training step mitigates to some extent the negative effects caused by an imperfect transfer set.

3. **(Optional) fine-tuning on** $\mathcal{D}_L$. This step makes the model robust to potential mismatches between the distribution of the transfer and labeled sets. We will refer to the two-step algorithm as PD, and to the three-step algorithm as PDF.

---

[1]Note that our $Transformer_{BASE}$ and $BERT_{BASE}$ in Devlin et al. (2018) have the same architecture. We use the former term for clarity, since not all students in Section 6 are pre-trained.

| Model | Step 1 ($\mathcal{D}_{LM}$) | Step 2 ($\mathcal{D}_T = \mathcal{D}_L$) | Architecture-agnostic |
|---|---|---|---|
| PD (our work) | LM pre-training | KD | ✓ |
| Sun et al. (2019a) | BERT$_{BASE}$ truncated | Patient-KD | ✗ |
| Sanh (2019) | BERT$_{BASE}$ truncated + LM-KD | Fine-tuning | ✗ |

Table 2: **Training Strategies** that build compact models by applying language model (LM) pre-training before knowledge distillation (KD). The first two rows apply distillation on task data. The third row applies distillation with an LM objective on general-domain data.

| | Model | SST-2 (acc) | MRPC (f1/acc) | QQP (f1/acc) | MNLI (acc m/mm) | QNLI (acc) | RTE (acc) | Meta Score |
|---|---|---|---|---|---|---|---|---|
| test | TF (baseline) | 90.7 | 85.9/80.2 | 69.2/88.2 | 80.4/79.7 | 86.7 | 63.6 | 80.5 |
| | PF (baseline) | **92.5** | **86.8/81.8** | 70.1/88.5 | 81.8/81.1 | 87.9 | 64.2 | 81.6 |
| | PD (our work) | 91.8 | **86.8**/81.7 | 70.4/**88.9** | **82.8/82.2** | 88.9 | 65.3 | **82.1** |
| | Sun et al. (2019a) | 92.0 | 85.0/79.9 | **70.7/88.9** | 81.5/81.0 | **89.0** | **65.5** | 81.7 |
| dev | PF (baseline) | 91.1 | 87.9/82.5 | 86.6/90.0 | 81.1/81.7 | 87.8 | 63.0 | 82.8 |
| | PD (our work) | 91.1 | **89.4/84.9** | 87.4/**90.7** | **82.5/83.4** | **89.4** | **66.7** | **84.4** |
| | Sanh (2019) | **92.7** | 88.3/82.4 | **87.7**/90.6 | 81.6/81.1 | 85.5 | 60.0 | 82.3 |

Table 3: **Model Quality.** All students are 6/768 BERT models, trained by 12/768 BERT teachers. Concurrent results are cited as reported by their authors. Our dev results are averaged over 5 runs. Our test results are evaluated on the GLUE server, using the model that performed best on dev. For anchoring, we also provide our results for MLM pre-training followed by fine-tuning (PF) and cite results from Sun et al. (2019a) for BERT$_{BASE}$ truncated and fine-tuned (TF). The meta score is computed on 6 tasks only, and is therefore not directly comparable to the GLUE leaderboard.

Figure 3: Pre-trained Distillation (PD) and concurrent work on model compression.

While we are treating our large teachers as black boxes, it is worth noting that they are produced by pre-training and fine-tuning. Since the teacher could potentially transfer the knowledge it has obtained via pre-training to the student through distillation, it is a priori unclear whether pre-training the student would bring additional benefits. As Section 6.2 shows, pre-training students is surprisingly important, even when millions of samples are available for transfer.

# 4 COMPARISON TO CONCURRENT WORK

There are concurrent efforts to ours aiming to leverage both pre-training and distillation in the context of building compact models. Though inspired by the two-stage pre-training+fine-tuning approach that enabled deep-and-wide architectures to advance the state-of-the-art in language understanding, they depart from this traditional method in several key ways.

Patient Knowledge Distillation (Sun et al., 2019a) initializes a student from the bottom layers of a deeper pre-trained model, then performs task-specific *patient* distillation. The training objective relies not only on the teacher output, but also on its intermediate layers, thus making assumptions about the student and teacher architectures. In a parallel line of work, DistilBert (Sanh, 2019) applies the same truncation-based initialization method for the student, then continues its LM pre-training via distillation from a more expensive LM teacher, and finally fine-tunes on task data. Its downside is that LM distillation is computationally expensive, as it requires a softmax operation over the entire vocabulary to compute the expensive LM teacher's predictive distribution. A common limitation in both studies is that the initialization strategy constrains the student to the teacher embedding size. Table 2 summarizes the differences between concurrent work and Pre-trained Distillation (PD).

To facilitate direct comparison, in this section we perform an experiment with the same model architecture, sizes and dataset settings used in the two studies mentioned above. We perform Pre-trained Distillation on a 6-layer BERT student with task supervision from a 12-layer BERT$_{BASE}$ teacher, using embedding size 768 for both models. For distillation, our transfer set coincides with

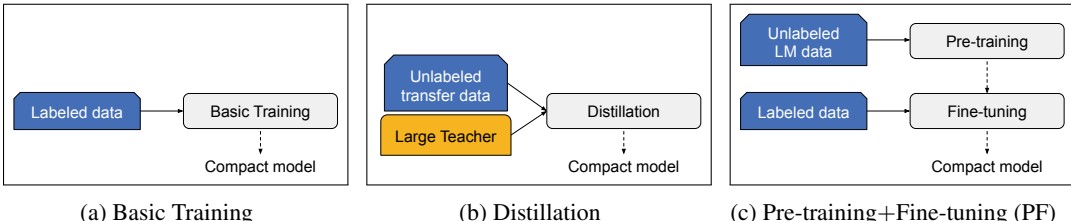

(a) Basic Training          (b) Distillation          (c) Pre-training+Fine-tuning (PF)

Figure 4: Baselines for building compact models, used for analysis (Section 6).

the labeled set ($\mathcal{D}_T = \mathcal{D}_L$). Table 3 reports results on the 6 GLUE tasks selected by Sun et al. (2019a) and shows that, on average, PD performs best. For anchoring, we also provide quality numbers for pre-training+fine-tuning (PF), which is surprisingly competitive to the more elaborate alternatives in this setting where $\mathcal{D}_T$ is not larger than $\mathcal{D}_L$. Remarkably, PF does not compromise generality or simplicity for quality. Its downside is, however, that it cannot leverage unlabeled task data and teacher model predictions.

## 5    ANALYSIS SETTINGS

Given these positive results, we aim to gain more insight into Pre-trained Distillation. We perform extensive analyses on two orthogonal axes—model sizes and properties of unlabeled data, thus departing from the settings used in Section 4.

All our models follow the Transformer architecture (Vaswani et al., 2017) and input processing used in BERT (Devlin et al., 2018). We denote the number of hidden layers as $L$ and the hidden embedding size as $H$, and refer to models by their $L/H$ dimensions. We always fix the number of self-attention heads to $H/64$ and the feed-forward/filter size to $4H$. The end-task models are obtained by stacking a linear classifier on top of the Transformer architectures.

The teacher, BERT$_{\text{LARGE}}$, has dimensions 24L/1024H and 340M parameters. We experiment with 24 student models, with sizes and relative latencies listed in Table 1. The most expensive student, Transformer$_{\text{BASE}}$, is 3 times smaller and 1.25 times faster than the teacher; the cheapest student, Transformer$_{\text{SMALL}}$, is 77 times smaller and 65 times faster. For readability, we report results on a selection of 5 students, but verify that all conclusions hold across the entire 24-model grid.

### 5.1    ANALYSIS BASELINES

We select three baselines for Pre-trained Distillation that can provide insights into the contributions made by each of its constituent operations.

**Basic Training** (Figure 4a) is the standard supervised learning method: a compact model is trained directly on the labeled set.

**Knowledge Distillation** (Figure 4b) (Bucilă et al., 2006; Hinton et al., 2015) (or simply "distillation") transfers information from a highly-parameterized and accurate *teacher* model to a more compact and thus less expressive *student*. For classification tasks, distillation exposes the student to *soft labels*, namely the class probabilities produced by the teacher $p_l = \text{softmax}(z_l/T)$, where $p_l$ is the output probability for class $l$, $z_l$ is the logit for class $l$, and $T$ is a constant called *temperature* that controls the smoothness of the output distribution. The softness of the labels enables better generalization than the gold *hard labels*. For each end task, we train: (*i*) **a teacher** obtained by fine-tuning pre-trained BERT$_{\text{LARGE}}$ (24L/1024H) on the labeled dataset (note teachers do not learn from the transfer set), and (*ii*) **24 students** of various sizes. Students are always distilled on the soft labels produced by the teacher with a temperature of $1^2$.

**Pre-training+Fine-tuning** (Figure 4c) (Dai & Le, 2015; Devlin et al., 2018), or simply PF, leverages large unlabeled general-domain corpora to pre-train models that can be fine-tuned for end tasks.

---

[2]While Hinton et al. (2015) show that tuning the temperature could increase performance, we did not observe notable gains. They also propose using a weighted sum of the soft and hard labels, but this approach cannot be applied directly in our set-up, since not all instances in our unlabeled transfers set have hard labels. Our optional final fine-tuning step similarly up-weights the hard labels in the labeled set.

Following BERT, we perform pre-training with the masked LM (MLM) and next sentence objectives (collectively referred to as $\text{MLM}^+$ from here on). The resulting model is fine-tuned on end-task labeled data. While pre-training large models has been shown to provide substantial benefits, we are unaware of any prior work systematically studying its effectiveness on compact architectures.

## 5.2 Analysis Tasks and Datasets

The tasks and associated datasets are summarized in Table 4.

**Sentiment classification** aims to classify text according to the polarities of opinions it contains. We perform 3-way document classification on Amazon Book Reviews (He & McAuley, 2016). Its considerable size (8m) allows us to closely follow the standard distillation setting, where there is a large number of unlabeled examples for transfer. Additionally, we test our algorithm on SST-2 (Socher et al., 2013),

| Labeled Data ($\mathcal{D}_L$) | Unlabeled Transfer Data ($\mathcal{D}_T$) |
| --- | --- |
| MNLI (390k) | NLI* (1.3m samples) |
| RTE (2.5k) | NLI* (1.3m samples) |
| SST-2 (68k) | Movie Reviews* (1.7m samples) |
| Book Reviews (50k) | Book Reviews* (8m samples) |

Table 4: **Datasets used for analysis.** A star indicates that hard labels are discarded. NLI* refers to the collection of MNLI (390k), SNLI (570k) and QQP (400k). The reviews datasets are part of Amazon Product Data. Unless otherwise stated, $\mathcal{D}_{LM}$ consists of BookCorpus & English Wikipedia.

which is a binary sentence classification task, and our results are directly comparable with prior work on the GLUE leaderboard (Wang et al., 2018). We use whole documents from Amazon Movie Reviews (1.7m) as unlabeled transfer data (note that SST-2 consists of single sentences).

**Natural language inference** involves classifying pairs of sentences (a premise and a hypothesis) as *entailment*, *contradiction*, or *neutral*. This task is representative of the scenario in which proxy data is non-trivial to gather (Gururangan et al., 2018). We chose MNLI (Williams et al., 2018) as our target dataset. Since strictly in-domain data is difficult to obtain, we supplement $\mathcal{D}_T$ with two other sentence-pair datasets: SNLI (Bowman et al., 2015) and QQP (Chen et al., 2018).

**Textual entailment** is similar to NLI, but restricted to binary classification (*entailment* vs *non-entailment*). The most popular RTE dataset (Bentivogli et al., 2009) is two orders of magnitude smaller than MNLI and offers an extreme test of robustness to the amount of transfer data.

## 6 Analysis

In this section, we conduct experiments that help us understand why Pre-trained Distillation is successful and how to attribute credit to its constituent operations.

### 6.1 There Are No Shortcuts: Why Full Pre-training Is Necessary

As later elaborated in Section 7, earlier efforts to leverage pre-training in the context of compact models simply feed pre-trained (possibly contextual) input representations into randomly-initialized students (Hu et al., 2018; Chia et al., 2018; Tang et al., 2019). Concurrent work initializes shallow-and-wide students from the bottom layers of their deeper pre-trained counterparts (Yang et al., 2019a; Sun et al., 2019a). The experiments below indicate these strategies are suboptimal, and that LM pre-training is necessary in order to unlock the full student potential.

**Is it enough to pre-train word embeddings?**     *No.* In order to prove that pre-training Transformer layers is important, we compare two flavors of Pre-trained Distillation[3]: PD with pre-trained word embeddings and PD with pre-trained word embeddings *and* Transformer layers. We produce word-piece embeddings by pre-training one-layer Transformers for each embedding size. We then discard the single Transformer layer and keep the embeddings to initialize our students.

For MNLI (Figure 5), less than 24% of the gains PD brings over distillation can be attributed to the pre-trained word embeddings (for Transformer$_{\text{TINY}}$, this drops even lower, to 5%). The rest of the benefits come from additionally pre-training the Transformer layers.

---

[3]Note that, for both flavors of PD, none of the student parameters are frozen; the word embeddings do get updated during distillation.

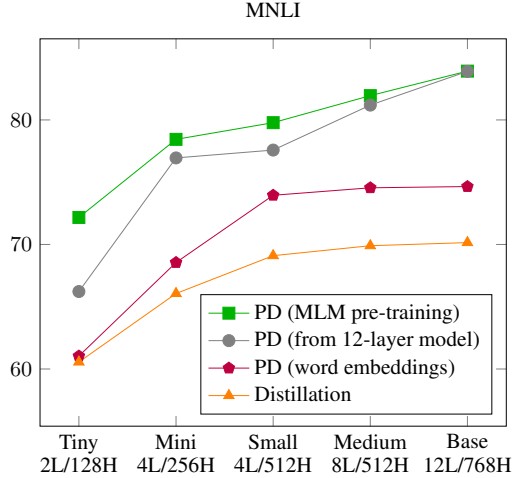
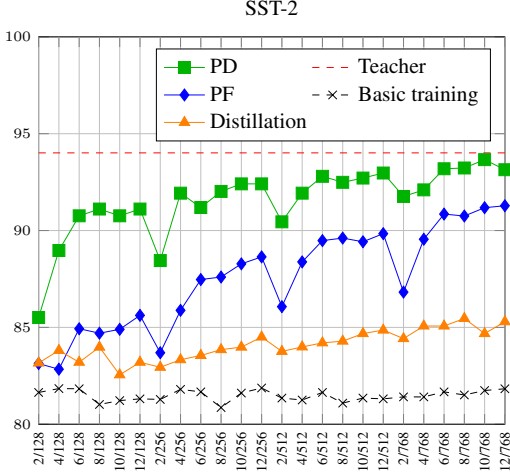

Figure 5: **Pre-training outperforms truncation.** Students initialized via LM pre-training (green) outperform those initialized from the bottom layers of 12-layer pre-trained models (gray). When only word embeddings are pre-trained (red), performance is degraded even further.

Figure 6: **Depth outweighs width** when models are pre-trained (PD and PF), as emphasized by the sharp drops in the plot. For instance, the 6L/512H (35.4m parameters) model outperforms the 2L/768H model (39.2m parameters). Randomly initialized models take poor advantage of extra parameters.

**Is it worse to truncate deep pre-trained models?** *Yes, especially for shallow students.* Given that pre-training is an expensive process, an exhaustive search over model sizes in the pursuit of the one that meets a certain performance threshold can be impractical. Instead of pre-training all (number of layers, embedding size) combinations of students, one way of short-cutting the process is to pre-train a single deep (e.g. 12-layer) student for each embedding size, then truncate it at various heights. Figure 5 shows that this can be detrimental especially to shallow architectures; Transformer_{TINY} loses more than 73% of the pre-training gains over distillation. As expected, losses fade away as the number of layers increases.

**What is the best student for a fixed parameter size budget?** *As a rule of thumb, prioritize depth over width, especially with pre-trained students.* Figure 6 presents a comparison between 24 student model architectures on SST-2, demonstrating how well different students utilize model capacity. They are sorted first by the hidden size, then by the number of layers. This roughly corresponds to a monotonic increase in the number of parameters, with a few exceptions for the largest students. The quality of randomly initialized students (i.e. basic training and distillation) is closely correlated with the number of parameters. With pre-training (i.e. PD and PF), we observe two intuitive findings: (1) pre-trained models are much more effective at using more parameters, and (2) pre-trained models are particularly efficient at utilizing depth, as indicated by the sharp drops in performance when moving to wider but shallower models.

This is yet another argument against initialization via truncation: for instance, truncating the bottom two layers of BERT_{BASE} would lead to a suboptimal distribution of parameters: the 2L/768H model (39.2m parameters) is dramatically worse than e.g. 6L/512H (35.4m parameters).

## 6.2 UNDER THE HOOD: DISSECTING PRE-TRAINED DISTILLATION

In the previous section, we presented empirical evidence for the importance of the initial LM pre-training step. In this section, we show that distillation brings additional value, especially in the presence of a considerably-sized transfer set, and that fine-tuning ensures robustness when the unlabeled data diverges from the labeled set.

**Comparison to analysis baselines** First, we quantify how much Pre-trained Distillation improves upon its constituent operations applied in isolation. We compare it against the baselines established in Section 5.1 (basic training, distillation, and pre-training+fine-tuning) on the three NLP tasks

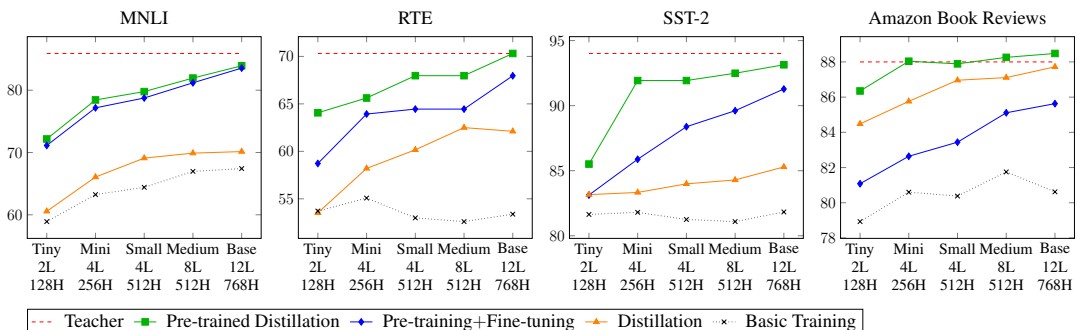

Figure 7: **Comparison against analysis baselines.** Pre-trained Distillation out-performs all baselines: pre-training+fine-tuning, distillation, and basic training over five different student sizes. Pre-training is performed on a large unlabeled LM set (BookCorpus & English Wikipedia). Distillation uses the task-specific unlabeled transfer sets listed in Table 4. Teachers are pre-trained BERT_{LARGE}, fine-tuned on labeled data.

described in Section 5.2. We use the BookCorpus (Zhu et al., 2015) and English Wikipedia as our unlabeled LM set, following the same pre-training procedure as Devlin et al. (2018).

Results in Figure 7 confirm that PD outperforms these baselines, with particularly remarkable results on the Amazon Book Reviews corpus, where Transformer_{MINI} recovers the accuracy of the teacher at a 31x decrease in model size and 16x speed-up. Distillation achieves the same performance with Transformer_{BASE}, which is 10x larger than Transformer_{MINI}. Thus PD can compress the model more effectively than distillation. On RTE, Pre-trained Distillation improves Transformer_{TINY} by more than 5% absolute over the closest baseline (pre-training+fine-tuning) and is the only method to recover teacher accuracy with Transformer_{BASE}.

It is interesting to note that the performance of the baseline systems is closely related to the size of the transfer set. For the sentence-pair tasks such as MNLI and RTE, where the size of the transfer set is moderate (1.3m) and slightly out-of-domain (see Table 4), pre-training+fine-tuning out-performs distillation across all student sizes, with an average of 12% for MNLI and 8% on RTE. Interestingly, the order is inverted on Amazon Book Reviews, where the large transfer set (8m) is strictly in-domain: distillation is better than pre-training+fine-tuning by an average of 3%. On the other hand, Pre-trained Distillation is consistently best in all cases. We will examine the robustness of Pre-trained Distillation in the rest of the section.

**Robustness to transfer set size** It is generally accepted that distillation is reliant upon a large transfer set. For instance, distillation for speech recognition is performed on hundreds of millions of data points (Li et al., 2014; Hinton et al., 2015).

We reaffirm this statement through experiments on Amazon Book Reviews in Figure 8, given that Amazon Book Reviews have the biggest transfer set. Distillation barely recovers teacher accuracy with the largest student (Transformer_{BASE}), using the entire 8m transfer set. When there is only 1m transfer set, the performance is 4% behind the teacher model. In contrast, PD achieves the same performance with Transformer_{MINI} on 5m instances. In other words, PD can match the teacher model with 10x smaller model and 1.5x less transfer data, compared to distillation.

**Robustness to domain shift** To the best of our knowledge, there is no prior work that explicitly studies how distillation is impacted by the mismatch between training and transfer sets (which we will refer to as *domain shift*). Many previous distillation efforts focus on tasks where the two sets come from the same distribution (Romero et al., 2014; Hinton et al., 2015), while others simply acknowledge the importance of and strive for a close match between them (Bucilǎ et al., 2006).

We provide empirical evidence that out-of-domain data degrades distillation and that our algorithm is more robust to mismatches between $\mathcal{D}_L$ and $\mathcal{D}_T$. We measure domain shift using the *Spearman rank correlation coefficient* (which we refer to as *Spearman* or simply *S*), introduced as a general metric in (Spearman, 1904) and first used as a corpus similarity metric in (Johansson et al., 1989). To compute corpus similarity, we follow the procedure described in (Kilgarriff & Rose, 1998): for two datasets $X$ and $Y$, we compute the corresponding frequency ranks $F_X$ and $F_Y$ of their most

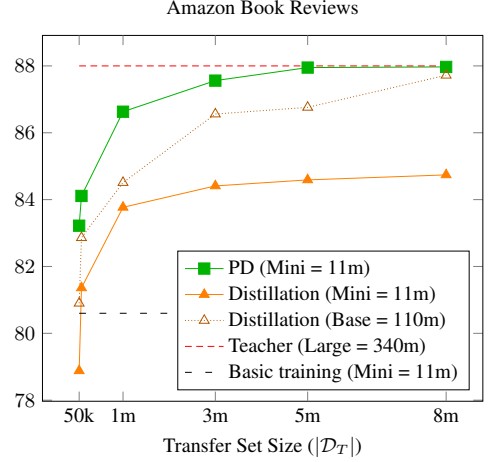

Figure 8: **Robustness to transfer set size.** We verify that distillation requires a *large transfer set*: 8m instances are needed to match the performance of the teacher using Transformer$_{\text{BASE}}$. PD achieves the same performance with Transformer$_{\text{MINI}}$, on a 5m transfer set (10x smaller, 13x faster, 1.5x less data).

Figure 9: **Robustness to domain shift in transfer set.** By keeping $|\mathcal{D}_T|$ fixed (1.7m) and varying the correlation between $\mathcal{D}_L$ and $\mathcal{D}_T$ (denoted by $S$), we show that distillation requires an *in-domain transfer set*. PD and PD-F are more robust to transfer set domain.

common $n = 100$ words. For each of these words, the difference $d$ between ranks in $F_X$ and $F_Y$ is computed. The final statistic is given by the following formula: $1 - \sum_{i=1}^{100} d_i^2/(n(n^2 - 1))$.

To measure the effect of domain shift, we again experiment on the Amazon Book Reviews task. Instead of varying the size of the transfer sets, this time we keep size fixed (to 1.7m documents) and vary the source of the unlabeled text used for distillation. Transfer set domains vary from not task-related (paragraphs from Wikipedia with $S$=0.43), to reviews for products of unrelated category (electronics reviews with $S$=0.52), followed by reviews from a related category (movie reviews with $S$=0.76), and finally in-domain book reviews ($S$=1.0). Results in Figure 9 show a direct correlation between accuracy and the Spearman coefficient for both distillation and PD. When $S$ drops to 0.43, distillation on $\mathcal{D}_T$ is 1.8% *worse* than basic training on $\mathcal{D}_L$, whereas PD suffers a smaller loss over pre-training+fine-tuning, and a gain of about 1.5% when a final fine-tuning step is added. When reviews from an unrelated product are used as a transfer set ($S$=0.52), PD obtains a much larger gain from learning from the teacher, compared to distillation.

## 6.3  BETTER TOGETHER: THE COMPOUND EFFECT OF PRE-TRAINING AND DISTILLATION

We investigate the interaction between pre-training and distillation by applying them sequentially on the *same* data. We compare the following two algorithms: Pre-training+Fine-tuning with $\mathcal{D}_{LM} = X$ and Pre-trained Distillation with $\mathcal{D}_{LM} = \mathcal{D}_T = X$. Any additional gains that the latter brings over the former must be attributed to distillation, providing evidence that the compound effect still exists.

For MNLI, we set $\mathcal{D}_{LM} = \mathcal{D}_T = \text{NLI*}$ and continue the experiment above by taking the students pre-trained on $\mathcal{D}_{LM} = \text{NLI*}$ and distilling them on $\mathcal{D}_T = \text{NLI*}$. As shown in Figure 10, PD is better than PF by 2.2% on average over all student sizes. Note that even when pre-training and then distilling on the *same data*, PD outperforms the two training strategies applied in isolation. The two methods are thus

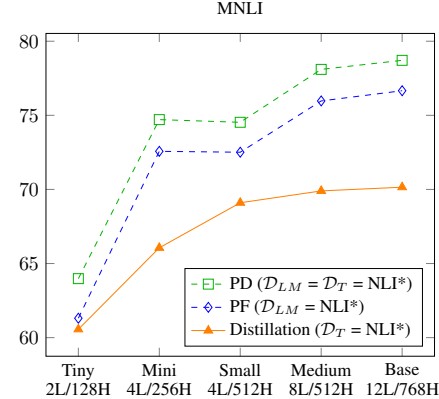

Figure 10: **Pre-training complements distillation.** PD outperforms the baselines even when we pre-train and distill on the same dataset ($\mathcal{D}_{LM} = \mathcal{D}_T = \text{NLI*}$).

learning different linguistic aspects, both useful for the end task.

## 7 RELATED WORK

**Pre-training** Decades of research have shown that unlabeled text can help learn language representations. Word embeddings were first used (Mikolov et al., 2013; Pennington et al., 2014), while subsequently contextual word representations were found more effective (Peters et al., 2018). Most recently, research has shifted towards *fine-tuning* methods (Radford et al., 2018; Devlin et al., 2018; Radford et al., 2019), where entire large pre-trained representations are fine-tuned for end tasks together with a small number of task-specific parameters. While feature-based unsupervised representations have been successfully used in compact models (Johnson & Zhang, 2015; Gururangan et al., 2019), *inter alia*, the pretraining+fine-tuning approach has not been studied in depth for such small models.

**Learning compact models** In this work we built on *model compression* (Bucilă et al., 2006) and its variant *knowledge distillation* (Hinton et al., 2015). Other related efforts introduced ways to transfer more information from a teacher to a student model, by sharing intermediate layer activations (Romero et al., 2014; Yim et al., 2017; Sun et al., 2019a). We experimented with related approaches, but found only slight gains which were dominated by the gains from pre-training and were not complementary. Prior works have also noted the unavailability of in-domain large-scale transfer data and proposed the use of automatically generated pseudo-examples (Bucilă et al., 2006; Kimura et al., 2018). Here we showed that large-scale general domain text can be successfully used for pre-training instead. A separate line of work uses pruning or quantization to derive smaller models (Han et al., 2016; Gupta et al., 2015). Gains from such techniques are expected to be complementary to PD.

**Distillation with unsupervised pre-training** Early efforts to leverage both unsupervised pre-training and distillation provide pre-trained (possibly contextual) word embeddings as inputs to students, rather than pre-training the student stack. For instance, Hu et al. (2018) use ELMo embeddings, while (Chia et al., 2018; Tang et al., 2019) use context-independent word embeddings. Concurrent work initializes Transformer students from the bottom layers of a 12-layer BERT model (Yang et al., 2019a; Sun et al., 2019a; Sanh, 2019). The latter continues student LM pre-training via distillation from a more expensive LM teacher. For a different purpose of deriving a single model for multiple tasks through distillation, Clark et al. (2019) use a pre-trained student model of the same size as multiple teacher models. However, none of the prior work has analyzed the impact of unsupervised learning for students in relation to the model size and domain of the transfer set.

## 8 CONCLUSION

We conducted extensive experiments to gain understanding of how knowledge distillation and the pre-training+fine-tuning algorithm work in isolation, and how they interact. We made the finding that their benefits compound, and unveiled the power of Pre-trained Distillation, a simple yet effective method to maximize the utilization of all available resources: a powerful teacher, and multiple sources of data (labeled sets, unlabeled transfer sets, and unlabeled LM sets).

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
