# OpenReview forum: "Well-Read Students Learn Better: On the Importance of Pre-training Compact Models"
_ICLR.cc/2020/Conference — Reject_

### Official Review · AnonReviewer1 · 2019-10-22
**Official Blind Review #1**

**Rating:** 1

**Review:**

The authors investigate the problem of training compact pre-trained language model via distillation. Their method consists of three steps:
1. pre-train the compact model LM
2. distill the compact model LM with a larger model (teacher)
3. fine-tune the compact model on target task

This idea is not significantly new since it is quite common to apply distillation to compress models, and the results are largely empirical. From Table 3 the results on test sets are better than previous works, but not by much. The authors spend quite a of space on ablation studies to investigate the contribution of different factors, and on cross-domain transfers. They do manage to show that using a teacher for distilling a compact student model does better than directly pre-training a compact model on the NLI* task in section 6.3. It would be better if they could show it for other tasks on the benchmark as well.

Overall I think this work is somewhat incremental, and falls below the acceptance threshold.


**Experience Assessment:**

I have published one or two papers in this area.

**Review Assessment: Checking Correctness Of Derivations And Theory:**

I assessed the sensibility of the derivations and theory.

**Review Assessment: Checking Correctness Of Experiments:**

I assessed the sensibility of the experiments.

**Review Assessment: Thoroughness In Paper Reading:**

I read the paper at least twice and used my best judgement in assessing the paper.

---

> ### Author Response · Authors · 2019-11-12
> **Response to official blind review #1**
>
> We believe the reviewer has misunderstood the contribution of the paper: our work does not present technical novelty, but an empirical demonstration that there has been significant overclaiming in the area where pre-training and distillation interact. In particular, multiple papers have advocated for highly restrictive yet complex strategies when more general, simpler baselines shown in our paper are just as effective.
>
> We argue that the ubiquity of distillation is not a strong enough reason to reject a paper that merely uses it as a tool. We do not claim novelty for using distillation in the context of building compact models, but rather investigate its interaction with pre-training. It was not clear a priori that a student with access to a powerful pre-trained teacher can (somewhat redundantly) benefit from its own pre-training. Prior cited studies initialize their model by truncating taller models, without questioning whether pre-training is necessary, or whether truncation is the best strategy for pre-training. Our in-depth ablation studies fill this void in the literature. Indeed, the results are empirical, not unlike the majority of neural network research.

---

### Official Review · AnonReviewer2 · 2019-10-22
**Official Blind Review #2**

**Rating:** 6

**Review:**

This submission revisits the student-teacher paradigm and shows through extensive experiments that pre-training a student directly on masked language modeling is better than distillation (from scratch). It also shows that the best is to combine both and distill from that pre-trained student model.

My rating is Weak Accept. I think the submission highlights a very useful observation about knowledge distillation that I imagine is overlooked by many researchers and practitioners. The decision of Weak as opposed to a Strong accept is because the submission does not introduce anything truly novel, but simply points out observations and offers a recommended training strategy. However, I do argue for its acceptance, because it does a thorough job and presents many interesting findings that can benefit the community.

Comparison with prior work:

The submission focuses on comparison with Sun et al. and Sanh. These comparisons are important, but not the most compelling part of the paper. Comparison with more prior work that show large benefits would make the paper even stronger.

Interesting experiments:

The paper presents many interesting experiments useful for anyone trying to develop a compressed model. First, it shows that distillation (from scratch) by itself may be overrated, since simply repeating the pre-training+fine-tuning procedure on the small model directly is effective. However, distillation remains relevant since it also shows that pre-training the student, then distilling against a teacher, is a potent combination. In the case when the transfer set is the same size as the pre-training set, it surprisingly still has some benefits. This is not experimentally explained, but I suspect there are optimization benefits that are hard to pin down exactly. The paper hypothesizes that the two methods learn different “linguistic aspects,” but I think it is a bit too speculative to put it in such terms.

The experiments are thorough, with many student sizes, transfer set sizes, transfer set/task set correlation, etc. It also compares against the truncation technique, where the student is initialized with a truncated version of the teacher. There are no error bars in the plots, but there are so many plots with clear trends, that this is not a big concern. I can’t think of any experiments that are obviously missing.

Misc:

- The introduction says that the pre-training+fine-tuning baseline has been overlooked. It would be great to point out papers that has actually overlooked this baseline. Including this in the results would be even better.
- During my first read-through, I got confused because I didn’t realize “pre-training” in most of the paper refers to “student pre-training” (as opposed to simply training the teacher). Making this a bit more explicit here and there can avoid this confusion.

**Experience Assessment:**

I have read many papers in this area.

**Review Assessment: Checking Correctness Of Derivations And Theory:**

I carefully checked the derivations and theory.

**Review Assessment: Checking Correctness Of Experiments:**

I assessed the sensibility of the experiments.

**Review Assessment: Thoroughness In Paper Reading:**

I read the paper thoroughly.

---

> ### Author Response · Authors · 2019-11-12
> **Response to official blind review #2**
>
> We thank the reviewer for taking the time to understand the subtleties of this problem, and reflect on how it can help the community.
>
> Regarding the misc comments:
> Our statement that pre-training+fine-tuning has been overlooked in prior work is exemplified by the two instances of prior work that we compared against (DistilBert and Patient Knowledge Distillation), which propose more elaborate methods without showing results for the simpler approach of directly applying the Bert recipe to smaller models.
> Acknowledged the confusion regarding which of the models is pre-trained; we are happy to clarify the wording in a future version.

---

### Official Review · AnonReviewer3 · 2019-10-23
**Official Blind Review #3**

**Rating:** 3

**Review:**

This paper proposes to pre-train a student before training with a teacher, which is easy to understand. Although the authors provide extensive empirical studies, I do not think they can justify the claims in this paper.


** Argument

One concern is that compared to other baselines such as "Patient knowledge distillation" [1], the proposed method is not consistently better. The authors argue that [1] is more sophisticated in that they distill task knowledge from intermediate teacher activations. However, the proposed method introduces other extra complexities, such as pre-training the student. I do not agree that the proposed method is less elaborate than previous methods.


Although the investigation on influence of model size and the amount/quality of unlabeled data is interesting in itself, this does not help justify the usefulness of pre-training a student. I hypothesize that when considering the intermediate feature maps as additional training signals, randomly initialized students can catch up with pre-trained students.

Furthermore, the mixed results shown in Table 3 do not justify the proposed method well enough.

[1] Patient Knowledge Distillation for BERT Model Compression, https://arxiv.org/abs/1908.09355

**Experience Assessment:**

I have published one or two papers in this area.

**Review Assessment: Checking Correctness Of Derivations And Theory:**

N/A

**Review Assessment: Checking Correctness Of Experiments:**

I assessed the sensibility of the experiments.

**Review Assessment: Thoroughness In Paper Reading:**

I read the paper at least twice and used my best judgement in assessing the paper.

---

> ### Author Response · Authors · 2019-11-12
> **Response to official blind review #3**
>
> We would like to reiterate the main take-aways of our paper, which are: 1) Pre-trained Distillation (PD) is an effective recipe in the specific setup where there is very little labeled data, but a more significant amount of task unlabeled data, and 2) PD is *just as good* as more elaborate techniques that make restrictive assumptions about the model architecture. We should have made it clearer that comparison against prior work in Table 3 is for completeness only, and it is not our goal to beat SoTA in the traditional setup; rather, we propose a solution for the case where there is unlabeled task data.
>
> Comparison to Patient Knowledge Distillation (PKD): PKD requires initialization with a pre-trained Transformer which has the same hidden dimension size and same or larger depth; it also requires that the teacher and student have the same hidden dimension size. It is possible that transferring intermediate map values will bring improvements on top of our method pre-trained distillation; this would be a possibility in the restricted case that the student and teacher have the same hidden dimension size, limiting the accuracy and efficiency of the compact model. Our preliminary experiments (not reported in the paper) showed no gains from intermediate map matching objectives when combined with pre-trained distillation but further experiments will be interesting.
>
> By eliminating the intermediate layer transfer between students and teachers, our method is more *general*, with no architectural restrictions. For instance, in Table 3, the comparison is limited to 6/768 models because of the extreme restrictions from the baselines. Also, we disagree that our method is more *elaborate* just because it requires a pre-training step; note that PKD also requires a (deeper) pre-trained Transformer. Once our pre-trained models are released, future efforts can simply reuse them, the same way PKD reuses a pre-trained Bert checkpoint. Exploring more flexible intermediate layer knowledge transfer following PKD but generalizing to mismatched dimensionality of student and teacher would be an interesting avenue for future work.

---

> > ### Comment · AnonReviewer3 · 2019-11-15
> > **Response**
> >
> > I read the authors' responses and am not satisfied.
> >
> > KD does not require that the teacher and student have the same hidden dimension size. This can be done following [FitNets: Hints for Thin Deep Nets](https://arxiv.org/pdf/1412.6550).

---

### Decision · Program_Chairs · 2019-12-19

**Decision:**

Reject

**Comment:**

 Though the reviewers thought the ideas in this paper were interesting, they questioned the importance and magnitude of the contribution.  Though it is important to share empirical results, the reviewers were not sure that there was enough for this paper to be accepted.